# Rapid antigen testing by community health workers for detection of SARS-CoV-2 in Dhaka, Bangladesh: a cross-sectional study

Ayesha Sania ,[1] Ahmed Nawsher Alam,[2] A S M Alamgir,[3,4] Joanna Andrecka,[5] Eric Brum,[5] Fergus Chadwick,[6] Tasnuva Chowdhury,[6] Zakiul Hasan,[5] Davina L Hill ,[6] Farzana Khan,[7] Mikolaj Kundegorski,[6] Seonjoo Lee,[8,9] Mahbubur Rahman ,[7] Yael K Rayport,[1,10] Tahmina Shirin,[2] Motahara Tasneem,[5] Katie Hampson[6]

For numbered affiliations see end of article.

**Correspondence to**
Dr Ayesha Sania;
as4823@cumc.columbia.edu

## ABSTRACT

**Objective** To evaluate the diagnostic performance and feasibility of rapid antigen testing for SARS-CoV-2 detection in low-income communities.

**Design** We conducted a cross-sectional community-based diagnostic accuracy study. Community health workers, who were trained and supervised by medical technicians, performed rapid antigen tests on symptomatic individuals, and up to two additional household members in their households and diagnostic results were calibrated against the gold standard RT-PCR.

**Setting** Low-income communities in Dhaka, Bangladesh.

**Participants** Between 19 May 2021 and 11 July 2021, 1240 nasal and saliva samples were collected from symptomatic individuals and 993 samples from additional household members (up to two from one household).

**Results** The sensitivity of rapid antigen tests was 0.68 on nasal samples (95% CI 0.62 to 0.73) and 0.41 on saliva (95% CI 0.35 to 0.46), with specificity also higher on nasal samples (0.98, 95% CI 0.97 to 0.99) than saliva (0.87, 95% CI 0.85 to 0.90). Testing up to two additional household members increased sensitivity to 0.71 on nasal samples (95% CI 0.65 to 0.76), but reduced specificity (0.96, 95% CI 0.94 to 0.97). Sensitivity on saliva rose to 0.48 (95% CI 0.42 to 0.54) with two additional household members tested but remained lower than sensitivity on nasal samples. During the study period, testing in these low-income communities increased fourfold through the mobilisation of community health workers for sample collection.

**Conclusions** Rapid antigen testing on nasal swabs can be effectively performed by community health workers yielding equivalent sensitivity and specificity to the literature. Household testing by community health workers in low-resource settings is an inexpensive approach that can increase testing capacity, accessibility and the effectiveness of control measures through immediately actionable results.

## INTRODUCTION

Since the start of the COVID-19 pandemic, countries have sought to develop rapid and

## STRENGTHS AND LIMITATIONS OF THIS STUDY

⇒ This study evaluates the performance of rapid antigen testing at the household level with all specimens collected by community health workers.

⇒ Sensitivity and specificity are reported for the index case and for testing of additional household members.

⇒ Lack of data on CT values precluded evaluation of the performance of rapid antigen tests according to viral load.

⇒ Information related to symptom duration was not available.

⇒ Diagnostic performance will depend not only on the proficiency of community health workers collecting specimens, but also the predominant SARS-CoV-2 variant(s) circulating and the incidence in the community.

accessible approaches to diagnostic testing. The capacity for Reverse Transcription Polymerase Chain Reaction (RT-PCR), the gold standard diagnostic for SARS-CoV-2, has ramped up around the world. Nonetheless, limitations remain in many low-resource settings because of the high cost, relatively slow turnaround time and need for specialised laboratory setup and highly skilled personnel to perform RT-PCR.[1–4] Rapid antigen tests are a promising alternative, as they are inexpensive, fast to perform (typically returning results within 15 min), and require only minimal technical skills and infrastructure.[5] Rapid antigen tests can therefore be used for symptomatic testing and as a screening tool to monitor the epidemiological situation. However, research is needed to evaluate the performance of rapid antigen tests and to develop an effective model for their deployment in many low-income and

middle-income country (LMIC) settings where adoption has been slow.

The diagnostic accuracy of rapid antigen tests has been evaluated in multiple clinical and laboratory studies, reporting high specificity and moderate to high sensitivity.[6–9] A recent meta-analysis of 133 clinical studies reported sensitivity of 76.3% (95% CI 73.1% to 79.2%), rising to 95.8% (95% CI 92.3% to 97.8%) when analyses were restricted to samples with high viral load (ie, CT values <25).[10] When the goal of testing is to mitigate transmission, the ideal test would be one that not just identifies the presence of SARS-CoV-2, but identifies those individuals that contribute to virus transmission.[11] Therefore, despite their lower sensitivity compared with the gold standard, and because of the immediacy of their results, as well as their low cost and ease of administration, rapid antigen tests are an ideal method for use in the community. Importantly, the direct administration of rapid antigen tests at households has the potential to lower testing costs and reduce other logistics barriers and risks, including the movement of infectious individuals when seeking tests. However, questions remain around the most appropriate sample type and delivery models for feasible and effective rapid antigen testing in communities.

Many countries that are constrained by limited diagnostic and healthcare infrastructure and human resources have used community health workers to aid service delivery during the pandemic.[12 13] In June 2020, responding to the Bangladesh Preparedness and Response Plan call to emergency action, the Food and Agriculture Organisation of the United Nations, the United Nations Populations Fund, and a collective of partner international agencies, non-governmental and civil society organisations (see the Acknowledgements section) initiated a community support team intervention under the leadership of the Bangladesh Government's Directorate General of Health Services. Community support teams comprised volunteers trained to search and screen symptomatic COVID-19 cases for preventing symptomatic transmission and were mobilised throughout the two city corporations of Dhaka, with an emphasis on socioeconomically deprived areas. Using this network, we evaluated the diagnostic performance and feasibility of rapid antigen testing for the identification of COVID-19 cases in the community. Here, we report on the sensitivity and specificity of rapid antigen testing by community workers on nasal and saliva samples from symptomatic individuals and their household members, including the impact of testing multiple household members on diagnostic performance. Based on these experiences in Dhaka, we further discuss the potential for community health workers to scale up rapid antigen testing for SARS-CoV-2 in low-income communities.

## METHODS
### Data collection
Community support teams comprised local volunteers that were trained by medical technologists to search for and identify suspected COVID-19 cases through hotline calls and community-based channels, and to counsel them in preventative measures, focusing on mask-wearing, isolation for suspected cases and quarantine of contacts for 14 days, as well as prescribing over-the-counter medications, and referring cases with severe symptoms to dedicated telemedicine services. At the time of the study, 750 volunteers, hereafter referred to as community health workers, within 465 community support teams were operating across under-represented vulnerable communities in Dhaka.

Between 19 May 2021 and 11 July 2021, study participants (n=1240) were recruited from low-income communities in Dhaka by a subset of community health workers (n=60, split into 30 teams of two persons) who had been trained in sample collection and testing by medical technologists. This subset of community health workers were selected by their line managers based on their performance within the community support teams, where they carry out syndromic surveillance to identify suspected COVID-19 cases and refer them for support. They primarily operated in 15 wards within both the Dhaka North and Dhaka South City Corporations, but sometimes travelled to neighbouring wards when potential COVID-19 cases were identified. At that time community health workers were primarily identifying cases through systematic searches within their communities. Community members aged 16 years or older with no previous medical history of bleeding disorders were considered eligible for the study.

Once the study began, index participants were considered eligible for recruitment if, on screening by the community health worker (guided by a mobile phone application), they were suspected to have COVID-19 infection because they reported with a fever (>38°C or >37.5°C following use of antipyretics) or a history of fever and had one or more of the following symptoms: difficulty breathing, cough, diarrhoea, headache, loss of smell, loss of taste, muscle pain, red eyes, runny nose, sore throat, tiredness, vomiting or a wet cough. Participants were then recruited if they consented to be tested. In each household, these suspected COVID-19 cases (index participants) were tested by the community health workers using rapid antigen tests, and then up to two additional household members, regardless of symptoms, were tested. These additional household members were recruited if they met the eligibility criteria (aged 16 or older and no history of bleeding disorder) and were available in the household at the time of testing, with those who had most contact with the index cases prioritised for selection. Community health workers uploaded participants' metadata, including their age, gender and symptoms as well as rapid antigen test results (see below), to a centralised database via the mobile application. Methods are described in further detail by Chadwick *et al.*[14]

From each participant, two mid-turbinate nasal swabs from both nostrils, as well as two oropharyngeal (OP) swabs, and raw saliva were collected. One nasal and

OP swab was pooled to create one sample for RT-PCR following protocols described by Vogels et al.[15] Combining nasal and OP swabs for RT-PCR was the standard guideline followed at Bangladesh's Institute of Epidemiology, Disease Control and Research (IEDCR). The swabs were sent to IEDCR for RT-PCR testing via cold storage, with samples for RT-PCR testing returned immediately using dedicated transportation. The nasal and saliva samples were tested independently with rapid antigen tests (SD Biosensor STANDARD Q COVID-19 Ag Test BioNote) in the household. Details of the sample collection and testing procedure described in online supplemental file 1.

In the pilot phase of the study, we evaluated the performance of OP, nasal and saliva swabs among symptomatic cases, which showed the performance of OP swabs slightly better than the nasal swabs. However, there were concerns of injury during OP swabs collected by the community health workers, so the rapid antigen tests were only performed on nasal and saliva swabs. The manufacturer reports that the STANDARD Q COVID-19 Ag Nasal Home Test has a sensitivity of 94.94% (75/79) and specificity of 100% (217/217)[16] and that the STANDARD Q COVID-19 Ag Saliva Home Test has a sensitivity of 94.74% (18/19) and a specificity of 100% (73/73).[17] In clinical settings, the nasal test has a lower sensitivity (66.7% to 100%), but high kappa values (0.92 to 0.852).[18] Similarly, the saliva test has a lower sensitivity (66.1%), but high specificity (99.6%) in clinical settings.[19]

All participants provided written informed consent for the health screening, collection of personal information and samples, and test results to be analysed in the study.

### Patient and public involvement

Participants were not involved in setting the research question or the outcome measures, but participants and their family members were intimately involved in the design and implementation of the study. During consultation in the design phase of the study, community members expressed their preference to be tested in their homes in preference to in a community gathering place. The index patient and their family members received their COVID-19 test results from their trusted community healthcare workers. They were also advised on isolation and quarantine measures, and they were connected with Government's telemedicine services when they needed.

### Statistical analysis

We assessed the distribution of patient age and sex by the numbers of participating household members using mean (SD) and counts (percentages) for continuous and categorical variables, respectively. We estimated test sensitivity specificity, positive predictive values (PPV), negative predictive values (NPV) and accuracy of rapid antigen tests with their 95% exact binomial CIs for all samples from index cases, according to sample type.[20] We further compared the sensitivity and specificity of the rapid antigen tests on each swab type and pooled, against RT-PCR, and results were further stratified by the numbers of additional household members tested. We compared group differences using Analysis of Variance (ANOVA) and $\chi^2$ tests as appropriate.

To examine how the use of community health workers could affect testing capacity we examined changes in the modality of sample collection and testing over time at IEDCR, corresponding to the training of community health workers for this study. Finally, we compared the expected number of positive COVID-19 diagnoses if RT-PCR testing was replaced with rapid antigen testing, based on our estimates of test sensitivity. For this hypothetical comparison we assume a constrained diagnostic testing budget and a cost of US$5 per rapid antigen test vs US$30 per RT-PCR laboratory diagnosis and we explored a range of prevalence levels within the community.

All analyses were undertaken in R Statistical Software (V.4.0.02)[21] and EpiR R package (V.2.0.39).[22] All data for the results presented in this paper is available in a repository accessible to the public.[23]

### RESULTS

During the study period, 1240 subjects consented to participate in the study. Of these subjects, 17 were excluded from the analyses due to missing information on the clinical definition used to diagnose a suspected case of COVID-19 (N=9, 0.7%), invalid results from rapid antigen tests on saliva samples or nasal swabs (4, 0.3%), or missing or invalid RT-PCR results (N=4, 0.3%). Table 1 shows the sex and age distribution of the participants by index person (N=1223), first additional household member (N=710) and second additional household member (N=283)

### Rapid antigen tests performed better on nasal than saliva samples

The PPV and NPV of rapid antigen tests on nasal swabs and saliva samples are shown in table 2 and figure 1 (red

| **Table 1** Demographic characteristics of the study participants | | | | | |
|---|---|---|---|---|---|
| | | **Index person (N=1223)** | **Second household member (N=710)** | **Third household member (N=283)** | **Total (N=2216)** |
| Sex* | Female | 575 (47.0%) | 397 (55.9%) | 151 (53.4%) | 1123 (50.7%) |
| | Male | 640 (52.3%) | 312 (43.9%) | 132 (46.6%) | 1084 (48.9%) |
| Age* | Mean (SD) | 36.522 (14.2) | 36.784 (14.42) | 34.735 (15.4) | 36.377 (14.48) |
| *Totals may not add up because of missing values. | | | | | |

| Table 2 | Performance of rapid antigen tests compared with RT-PCR, among all index household members tested | |
|---|---|---|
| | Nasal sample | Saliva sample |
| | PCR + PCR − total | PCR + PCR − total |
| | RAT + 198 21 219 | RAT + 118 117 235 |
| | RAT − 93 911 1004 | RAT − 173 815 988 |
| | Total 291 932 1223 | Total 291 932 1223 |
| Sensitivity | 0.68 (0.62 to 0.73) | 0.41 (0.35 to 0.46) |
| Specificity | 0.98 (0.97 to 0.99) | 0.87 (0.85 to 0.90) |
| PPV | 0.90 (0.86 to 0.94) | 0.50 (0.44 to 0.57) |
| NPV | 0.91 (0.89 to 0.92) | 0.82 (0.80 to 0.85) |
| Accuracy | 0.91 (0.89 to 0.92) | 0.76 (0.74 to 0.79) |

The total sample size was 1223 index subjects, 95% CIs are shown in brackets.
NPV, negative predictive value; PPV, positive predictive value; RAT, rapid antigen test; RT-PCR, Reverse transcription polymerase chain reaction .

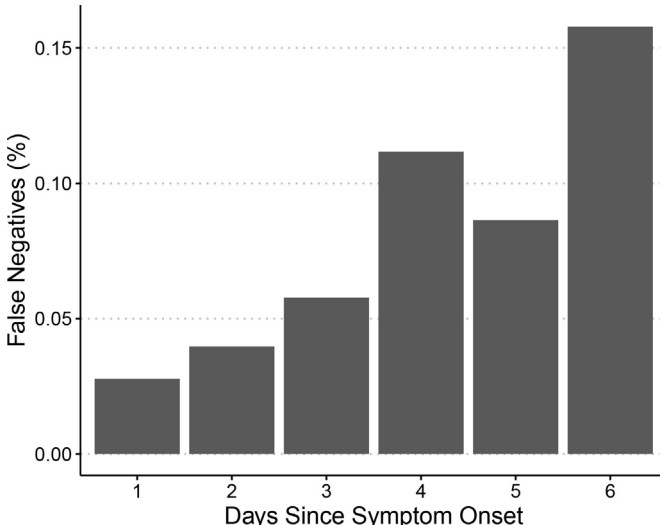

**Figure 2** The percentage of false negative rapid antigen test results given a positive RT-PCR increases with days from self-reported symptom onset. RT-PCR, Reverse Transcription Polymerase Chain Reaction.

bars). Of all subjects with complete test results (N=1223), 291 were RT-PCR positive on pooled OP and nasal samples. Of these, 198 were rapid antigen test positive by nasal swab and 118 were rapid antigen test positive by saliva only sample. Therefore, the overall sensitivity of rapid antigen testing on nasal samples was 0.68 (95% CI 0.62 to 0.73) and on saliva samples was 0.41 (95% CI 0.35 to 0.46). Saliva samples picked up some RT-PCR-positive subjects who were not positive in the nasal swab (n=22), and vice versa (n=102). Specificity of nasal swabs (0.98, 95% CI 0.97 to 0.99) was also higher than saliva samples (0.87, 95% CI 0.85 to 0.90). The overall accuracy of rapid antigen tests on nasal swabs was higher (0.91 vs 0.76) compared with accuracy on saliva samples (table 2).

The probability of a false negative rapid antigen test (ie, discordant with a positive RT-PCR test) increased with the number of days between self-reported symptom onset and testing, and very few false negatives occurred

within the first 3 days of symptoms (figure 2). However, our sample size (55 false negative rapid antigen tests from 736 test results with self-reported symptoms onset date) was too small for rigorous quantitative analysis of this relationship.

### Testing additional household members only marginally increases sensitivity but decreases specificity

The sensitivity of rapid antigen tests on nasal swabs increased from 0.68 (95% CI 0.62 to 0.73) when only the index person was tested, to 0.70 (95% CI 0.65 to 0.76) when an additional household member was tested (figure 1, green bars) and to 0.71 (95% CI 0.65 to 0.76) when a second household member was also tested (figure 1, blue bars), although it should be noted that only a subset of households had additional members available for testing at the time of community health worker visits. With this

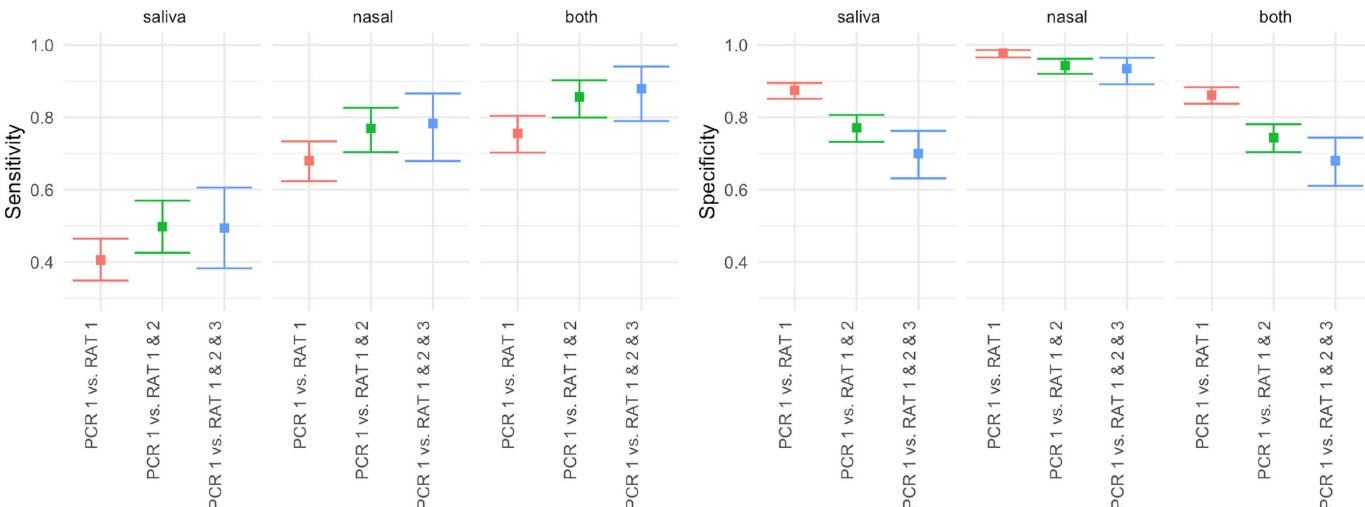

**Figure 1** Sensitivity (left) and specificity (right) of rapid antigen testing compared with RT-PCR, when 1 (index person), 2 or 3 members of the same household are tested. RT-PCR, Reverse Transcription Polymerase Chain Reaction.

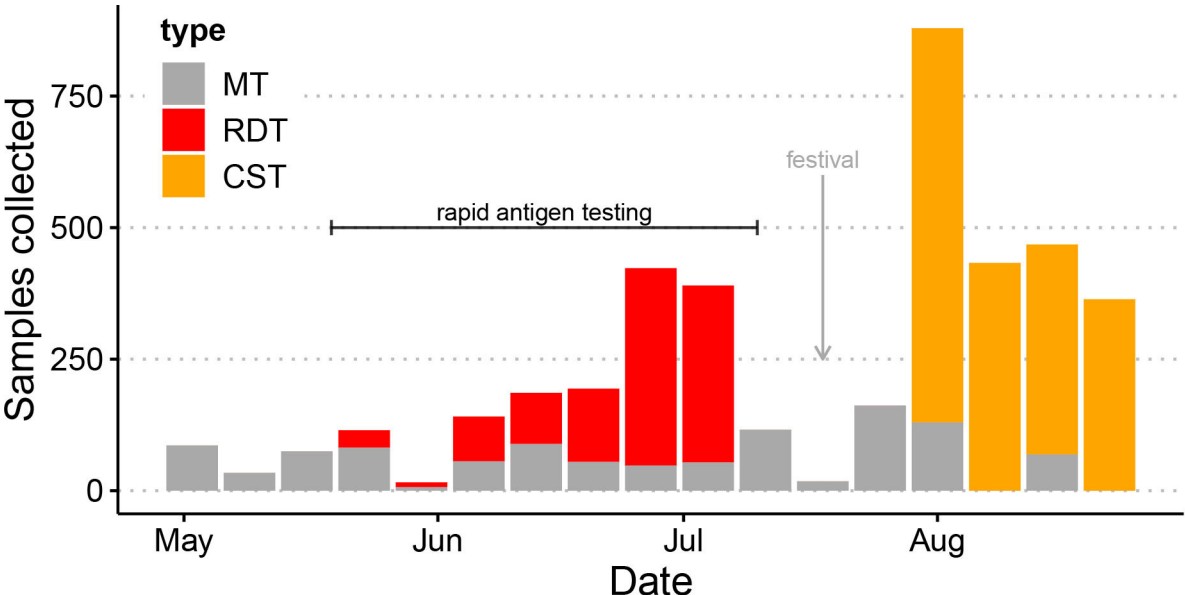

**Figure 3** IEDCR sample collection and testing in Dhaka from May to September 2021. Routine sample collection by IEDCR from low-income communities in Dhaka where the community support teams operate. Sample collections by medical technologists (MT)are shown in grey. The period of rapid antigen testing is indicated, with samples collected for rapid testing in households shown in red (RDT). The subsequent collection of samples by community health workers for RT-PCR testing during the surge of infection driven by the Delta variant is shown in orange. A marked dip in testing in late July corresponded to a religious festival (Eid Al-Adha). IEDCR, Institute of Epidemiology, Disease Control and Research; RDT, rapid antigen test. RT-PCR, Reverse Transcription Polymerase Chain Reaction. CST, Community Support Teams

gain in sensitivity, there was a corresponding drop in specificity with additional household members, from 0.98 (95% CI 0.97 to 099) to 0.96 (95% CI 0.94 to 0.97) for two additional members tested (figure 1). Similarly, sensitivity of rapid antigen testing on saliva samples increased from 0.42 (95% CI 0.35 to 0.46) to 0.48 (95% CI 0.42 to 0.54) with two additional household members tested, although there was a corresponding drop in specificity from 0.87 (95%CI 0.85 to 0.90) to 0.82 (95% CI 0.79 to 0.84). Testing both sample types also yielded a modest increase in sensitivity but a large drop in specificity (figure 1).

### Community health workers achieved higher testing coverage than routine laboratory testing coverage

During the validation phase of this study between 19 May 2021 and 11 July 2021, community health workers collected 73% of the samples from these low-income communities that were tested at IEDCR (figure 3). Based on the increase in testing volume in these communities and the self-reported experiences of the community health workers involved in this work, we deduced that testing at household level promotes testing. Indeed, the willingness of symptomatic individuals to be tested more than doubled, when testing was offered in people's homes by the community health workers, in comparison to when community health workers offered free testing the next day by medical technologists, which was also in people's homes or at designated local test points.

Shortly after the study began, the Delta variant swept across Bangladesh, and by late June community workers were collecting around 90% of samples in these communities as case numbers surged. The community health workers trained to collect samples for this study continued to collect samples for IEDCR as part of their routine RT-PCR surveillance during this third wave of infection (figure 3). This period of implementation demonstrated that community health workers can collect and test over four times more households compared with the standard approach of community testing (figure 3), which relies on follow-up visits by medical technologists the day after symptomatic households are identified by community health workers. Assuming a constrained diagnostic testing budget and a cost of US$5 per rapid diagnostic test and of US$30 per RT-PCR laboratory diagnosis, we calculate that household (rapid antigen) testing would detect four times more cases than laboratory (RT-PCR) testing as shown in figure 4, despite its lower sensitivity, under a range of prevalence levels (from <0.05 to >0.4, ie, spanning the range of test positivity reported from Dhaka in 2021).

### DISCUSSION

We evaluated the performance of rapid antigen testing on nasal and saliva samples collected by community health workers at the household level in low-income communities across Dhaka, Bangladesh. We estimated that rapid antigen tests on nasal swabs had moderate sensitivity and high specificity as expected in a community setting. We also found that testing additional household members increased the sensitivity of detection of

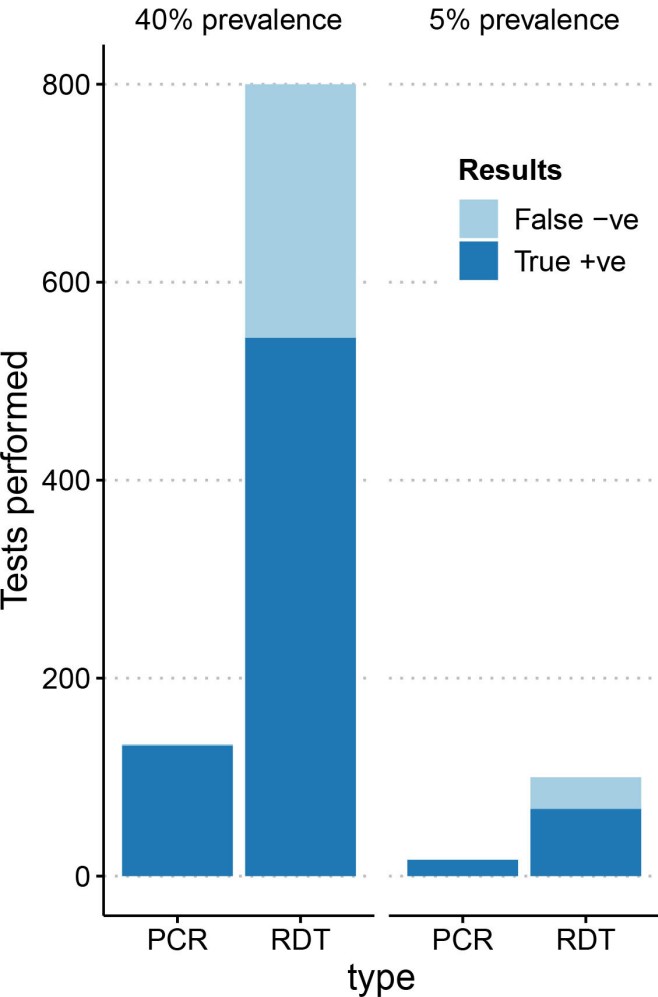

**Figure 4** Comparison of positive cases confirmed by rapid antigen testing versus RT-PCR under a specified diagnostic testing budget. We assume that the per cost test of rapid antigen tests (RDT) is US$5 and of RT-PCR is US$30, and the overall testing budget is US$10 000. Modelled results are shown for prevalence of 40% (test positivity exceeded 60% at the peak of the Beta and Delta waves) and of 5% (test positivity rarely dropped below 5% in 2021). Test results are shown based on the estimated rapid antigen test sensitivity and specificity that we reported for testing index cases. RT-PCR, Reverse Transcription Polymerase Chain Reaction.

COVID-19 infection only marginally, but decreased specificity. This study capitalised on a network of community volunteers who were initially trained to clinically assess individuals reporting COVID-19 symptoms as part of the crisis response and demonstrated that community health workers can administer rapid antigen tests effectively and achieve high coverage in community settings. Our results support recommendations for scaling up rapid testing as part of the COVID-19 response in low-resource settings.

Our report of the superior performance of rapid antigen tests on nasal samples compared with saliva samples confirms the existing literature from clinical and laboratory settings.[24] A recent study reported poor concordance between nasopharyngeal swab specimens and saliva samples (75.9%, kappa coefficient 0.310).[25]

However, a meta-analysis examining the performance of RT-PCR concluded that saliva samples were as sensitive as nasopharyngeal swabs.[26 27] A likely explanation for the lower sensitivity of rapid antigen tests on saliva could be due to lower viral load, which would have less impact with RT-PCR.[28] A study on the diagnostic accuracy of rapid antigen tests conducted at testing sites in the Netherlands found that their sensitivity increased from 60% to 85% after applying a viral load cut-off as a proxy for infectiousness.[29] The performance of rapid antigen tests further depends on the virus variant, with a higher viral load reported in saliva for the Beta variant compared with others.[30] Although saliva samples have advantages including easier administration (typically self-collected for testing in many high-income settings), being less invasive, and causing minimal discomfort,[31] we confirm the superiority of nasal swabs in terms of sensitivity and specificity, at least for the Delta variant, which was most prevalent in Dhaka during our study.[32]

Our estimate of rapid antigen test sensitivity on nasal samples (68%) is comparable to that reported by Cerruti *et al* (70.6 %) from a mix of symptomatic and asymptomatic participants.[33] Estimates of sensitivity reported by the manufacturer and by studies conducted among symptomatic patients at the point of care are, however, much higher than in our report. SD biosensor, the manufacturer of the STANDARD Q COVID-19 Ag Nasal Home Test reports a sensitivity of 94.94%,[16] and an SD biosensor rapid antigen test had a sensitivity of 89.0% at a point-of-care community-based testing centre, where the vast majority of the patients were symptomatic (97.8%).[34] Higher sensitivity in laboratory and clinical settings is likely due to higher viral loads. Several prior reports including a recent meta-analysis confirmed that rapid antigen tests have their highest sensitivity in samples with high viral load (CT value <25)[10] and therefore lower sensitivity is to be expected when testing in the community includes both individuals who are currently infectious as well as those who are recovering (figure 2).[11] Moreover, a point-of-care rapid antigen test study found that when rapid antigen tests and RT-PCR results were discordant, SARS-CoV-2 could not be cultured from the samples, suggesting that these patients were unlikely to be infectious.[35] Thus, from a public health perspective, where curtailing disease transmission is the goal, only people with high viral loads need to be detected as they are capable of transmitting the disease to others.[4 36] Rapid antigen tests are better suited to screening for people with high viral loads, the population which poses the greatest risk for the community. This combined with their reduced costs, logistics and speed of turnaround makes them advantageous for supporting the rapid and effective implementation of targeted prevention measures.

To the best of our knowledge, this is the first study to report on the performance of rapid antigen testing at the household level. The hypothesis that testing of additional household members could improve sensitivity was motivated by the previous report of higher sensitivity achieved

in flock-based testing in animals.[37] For highly pathogenic avian influenza H5N1 in backyard poultry flocks, Robyn et al[38] demonstrated that combining a standard case definition with one or more rapid antigen tests can increase the sensitivity of case detection to closer to that of the gold standard RT-PCR while maintaining specificity. Depending on the prevalence of COVID-19 cases in the community, testing additional household members may slightly improve sensitivity. However, this strategy will incur higher costs as well as a greater workload for the community health workers. Overall, our findings underscore the importance of testing symptomatic patients in the community.

A limitation of our study is the lack of data on CT values which precluded evaluation of the performance of rapid antigen tests according to viral load (with CT values as a proxy). The higher sensitivity and specificity in samples with high viral load have been confirmed in several prior studies.[4 11] Recent reports indicate that a large proportion of asymptomatic COVID-19 cases also have high viral load.[39 40] Since the eligibility criteria to be an index participant in our study included presence of fever and other covid-like symptoms, the estimates of sensitivity and specificity from our study are not generalisable to asymptomatic cases. Another limitation is that while we found that the probability of false negative results increases as the duration of symptoms increases, our sample size (55 false negative rapid antigen tests from 736 test results with self-reported symptoms onset date) was too small for rigorous quantitative analysis of this relationship. Finally, our study was conducted during a period of high COVID-19 transmission in the community and the index participants were symptomatic, indicating a high pretest probability, which impacts the PPV and NPV of a test. Therefore, the PPV and NPV in other settings will differ depending on the pretest probability, and the accuracy of rapid tests will also vary with the quality of the training provided to community health workers and their supervision.

Our study was conducted in a low-resource community setting among a large sample of subjects (n=1240) with all specimens collected by community health workers. These communities are underserved by routine testing procedures in Bangladesh, which require that individuals visit a testing facility and pay for a test (US$15–US$50 for a RT-PCR with results returned 2–5 days later and more recently US$5–US$10 for a rapid antigen test). During the peak of the pandemic, to access care for complications due to COVID-19 patients were required to present a positive test result, and those requiring non-COVID-19 care were required to present with a negative test, affecting the care that individuals could access. Community health workers helped to improve a critical gap in access to testing, which increased the number of people tested from these communities. Moreover, people in these communities were keen to be tested as results were returned immediately in the privacy of their own household. Additionally, familiarity and trust of the community health workers likely contributed to people's willingness to be tested. During the rapid antigen testing validation, the Delta variant began to surge in Dhaka, and community health workers trained to collect samples as part of this study were deployed to continue sample collection for RT-PCR testing. In August 2021 community support teams supported by Food and Agriculture Organization (FAO) and United Nations Population Fund (UNFPA) collected over 80% of the samples from these communities that were tested at the national laboratory, IEDCR, during Bangladesh's third wave, which was the largest peak experienced so far. Sample collection by community health workers proved more efficient and acceptable in these communities and meant that IEDCR was better able to meet testing demand at this critical time.

During the COVID-19 pandemic, many LMIC governments have established community health worker programmes to support diagnostic testing, treatment, immunisation and contact tracing.[41] Global initiatives have sought to reduce the costs of rapid antigen tests for COVID-19 to below US$5 each, with catalytic funding to guarantee prices for LMICs.[12 27] Our study demonstrates that community health workers are capable of effective household-level rapid antigen testing in low-income communities in Dhaka. This provides an example that can be replicated in many community-based programmes in Bangladesh and other low-resource settings. Future research should evaluate the scalability of rapid testing deployed by community health workers to mitigate COVID-19 in LMIC communities.

**Author affiliations**

[1]Division of Developmental Neuroscience, Department of Psychiatry, Columbia University Irving Medical Center, New York, New York, USA
[2]Department of Virology, Institute of Epidemiology Disease Control and Research, Dhaka, Bangladesh
[3]Department of Entomology, Institute of Epidemiology Disease Control and Research, Dhaka, Bangladesh
[4]Centre for Food and Waterborne Diseases, ICDDR,B, Dhaka, Bangladesh
[5]Food and Agriculture Organization of the United Nations, Dhaka, Bangladesh
[6]University of Glasgow Institute of Biodiversity Animal Health and Comparative Medicine, Glasgow, UK
[7]Department of Epidemiology, Institute of Epidemiology Disease Control and Research, Dhaka, Bangladesh
[8]Division of Mental Health Data Science, New York State Psychiatric Institute, New York, New York, USA
[9]Department of Biostatistics, Columbia University Mailman School of Public Health, New York, New York, USA
[10]Department of Neuroscience, New York State Psychiatric Institute, New York, New York, USA

**Acknowledgements** The authors thank the Directorate General of Health Services (DGHS), Bangladesh, and the Institute of Epidemiology Disease Control and Research (IEDCR) for ongoing SAR-CoV-2 surveillance and prevention measures. The Community Support Team intervention is funded by the World Bank, USAID, and FCDO (UKAID). This multisectoral, collaborative initiative is led by the DGHS with technical and operational support from several UN Organisations (FAO, UNFPA, UNICEF, WFP), with community health workers and volunteers provided by NGOs (BRAC) and civil society organisations (Youth Platform, Utshargo Foundation, Himu Paribahan, CDP, Platform and Young Bangla). Additional technical support was provided by national institutions such as a2i and icddr,b. We are extremely grateful to the community support team workers who have volunteered throughout this emergency, dedicating time and energy to provide support and response in these

communities and without whom this study would not have been possible, as well as to the Zonal Executive Officers, Ward Councilors, Chief Health Officers, Deputy Chief Health Officers and other administrative members of Dhaka North and Dhaka South City Corporations. The University of Glasgow LMICs COVID-19 working group provided valuable feedback and discussion.

**Contributors** EB, AS and KH conceptualised and designed the study, and EB secured funding. AS, EB, TS, ASMA, MR, ANA, FK and KH planned the study and provided scientific, administrative and technical support during implementation and data collection of the study. TC developed the standard operating procedures for collection of the samples from the community. MK developed the data collection tools and managed the data. JA coordinated activities, contributed in data analysis, interpretation of results and drafting of the manuscript. MT led the field implementation, and with ZH trained and supervised the community health workers and provided technical support during data collection. AS wrote the data analysis plan, and the first manuscript draft. SL conducted data analysis, contributed in interpretation of results and revised the manuscript. YKR assisted in literature reviews and manuscript drafting. KH, FC and DLH undertook the data analysis, data interpretation and manuscript revisions. EB, KH and AS had full access to all of the data in the study and took responsibility for the integrity of the data and the accuracy of the data analysis. All authors reviewed and approved the final manuscript. AS, EB, and KH accepts full responsibility for the work and/or the conduct of the study, had access to the data, and controlled the decision to publish.

**Funding** The Bill & Melinda Gates Foundation (BMGF) funded work by FAO (INV-022851) and UoG reports funding from Wellcome (207569/Z/17/Z to KH).

**Competing interests** None declared.

**Patient and public involvement** Patients and/or the public were involved in the design, or conduct, or reporting, or dissemination plans of this research. Refer to the Methods section for further details.

**Patient consent for publication** Consent obtained from parent(s)/guardian(s)

**Ethics approval** This study involves human participants and was approved by Institutional Review Board at the IEDCR, Ministry of Health, Bangladesh, IEDCR/IRB/2021/04.

**Provenance and peer review** Not commissioned; externally peer reviewed.

**Data availability statement** Data are available in a public, open access repository. Data and code for replication of the study results are available from: https://zenodo.org/badge/latestdoi/432151026.

**ORCID iDs**
Ayesha Sania http://orcid.org/0000-0002-6445-9481
Davina L Hill http://orcid.org/0000-0001-9085-6192
Mahbubur Rahman http://orcid.org/0000-0001-8577-8281

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
