## [Reviewer comments · BMJ Open]

ARTICLE DETAILS

TITLE (PROVISIONAL)	Rapid antigen testing by community health workers for detection of SARS-CoV-2 in Dhaka, Bangladesh: a cross-sectional study
AUTHORS	Sania, Ayesha; Alam, Ahmed; Alamgir, A. S. M; Andrecka, Joanna; Brum, Eric; Chadwick, Fergus; Chowdhury, Tasnuva; Hasan, Md. Zakiul; Hill, Davina; Khan, Farzana; Kundegorski, Mikolaj; Lee, Seonjoo; Rahman, Mahbubur; Rayport, Yael; Shirin, Tahmina; Tasneem, Motahara; Hampson, Katie

VERSION 1 – REVIEW

REVIEWER	Isah Abubakar Aliyu Bayero University
REVIEW RETURNED	08-Feb-2022

GENERAL COMMENTS	The research work by Sania et al. titled Rapid antigen testing by community health workers to improve detection of SARA-CoV-2 in Dhaka, Bangladesh submitted to BMJ Open, aimed at evaluating the diagnostic performance and feasibility of rapid antigen testing for SARS-CoV-2 detection by community support team in low-income communities. though the paper was well written and authors attempts to address the objectives of their research, i have the following observation to make the paper better. 1) Methods; it was not clear how the household were selected for their research, how these household members were mobilized, it was also not clear which part of Dhaka was sampled. they did mention low income communities only. 2) Result; on the page 10 of 23, line 8: " Table 1 shows the sex and age distribution of the participants...." this statement sound inappropriate, it is their result rather then the table. it should ideally beour result showed that/the finding of this study showed....etc (Table 1). The tables should be presented in a more scientific manner. Page 10 of 23 line 36,; same result was presented in table and figure, the authors should chose either to represent same group of data. conclusion; the bases with which the authors recommend using community health worker to do laboratory testing was not justified by this research.
--

REVIEWER	Lisa Bebell Massachusetts General Hospital
REVIEW RETURNED	08-Mar-2022

GENERAL COMMENTS	The authors present their findings on an important research topic – testing for SARS-CoV-2 by community support teams in Bangladesh. This is an important study to help inform how rapid
--

testing can expand access to COVID-19 diagnostics, and support the role of CHWs. However, I think that prior to publishing this work, additional detail is needed in the Methods and Results sections. Furthermore, I disagree with the interpretation of the change in sensitivity and specificity that occurred when additional household members were tested. I think these results should be reconsidered and the Discussion section reframed.

Specific comments:

Abstract:

I recommend updating/reframing the abstract to reflect my comments below for each individual section.

Introduction:

Appropriate to the topic and flows well. The introduction could be shortened slightly if word count is of concern – especially the last paragraph, which discusses healthcare worker deployment and testing in greater detail than is necessary in the introduction.

Methods:

Additional clarity is needed in the methods section to help the reader understand exactly how CHWs and participants were selected, and what samples were collected from each person for each type of test – the results section should also be updated to parallel what is delineated in the methods.

Page 8, Lines 20-24: Please clarify how ‘participants’ were chosen. There is some clarification provided in the following sentences (noting that CHWs were identifying symptomatic cases) – were all participants suspected cases? How were the smaller CHW groups (n=60, 30 teams) selected? Did they operate in a single geographic area, or in all communities?

Page 8, Lines 26-27: The authors note “Participants aged 16 years or older with no previous medical history of bleeding disorders were considered eligible for the study.” Do the authors mean community members? How would they be participants if they hadn’t already been consented for the study? Or were they already consented for another study? Why were people with bleeding disorders excluded? Were household members only eligible if they were also over 16 years of age?

Page 8, Line 46: The authors write “nasal swabs from each nostril” – does this mean one swab from each nostril? Were these swabs of the anterior nares? Some of these details are important for correctly interpreting the results.

Page 8, Line 47: When the authors write “One nasal and OP swab was pooled for RT-PCR” do they mean that the nasal and OP swab were combined to create one sample for testing?

Page 8, Lines 51-52: The authors write “The nasal and saliva samples were tested independently with rapid antigen tests” Does this mean that no OP swabs were tested with rapid antigen tests? And that each nare was tested separately with a rapid antigen test for each nare?

Page 8-9: The authors write “The manufacturer reports that the STANDARD Q COVID-19 Ag Nasal Home Test has a sensitivity of 94.94% (75/79) and specificity of 100% (217/217)(16) and that the STANDARD Q COVID-19 Ag Saliva Home Test has a sensitivity of 94.74% (18/19) and a specificity of 100% (73/73).(17)” Was this test truly validated on this few (<300) samples? Are there real-world results reporting the field sensitivity and specificity? I ask because

	the reported sensitivity and specificity seem to high to be realistic. Page 9, Lines 14-21: This paragraph describes patient involvement. Who were the patients? Were they the same as the participants? I find this confusing – please clarify whether this describes participant involvement or patients who were not study participants. Also, how were patients and their family members involved with the study design? Page 9, Lines 31-34: Do the authors mean that they studied the potential cost savings of substituting antigen for PCR testing? Please clarify. Results: This section does not parallel the description in the Methods section. Additional detail in the Methods section will help – the Results should then describe all the tests discussed in the Methods section. As it is written now, the Results section does not mirror the Methods section well, especially when describing the pooled versus non-pooled testing results and nasal versus OP versus saliva testing results. Page 10, Lines 5-6: How many potential participants declined to participate? Please clarify the results reporting saliva samples – are these saliva only, or OP samples, or pooled samples? Page 11, Lines 43-45: This sentence seems like it would be better placed in the Discussion. Page 11: The authors begin a section with the heading “Testing additional household members yields better rapid antigen test performance.” However, this paragraph describes essentially no change in sensitivity of nasal testing by adding 1-2 household contacts (an increase of 0.68 to 0.71 does not seem meaningful) and a decrement in specificity for both nasal and saliva test performance. I would not call this ‘better rapid antigen test performance’. Thus, I recommend re-heading this section, perhaps with “Testing additional household members decreases rapid antigen test specificity” to ensure the message is clear. Page 12, Lines 16-19: “Using the rapid antigen testing approach, whereby individuals were tested in their households and immediately given their results, willingness to be tested greatly increased.” I believe this statement is true – can the authors provide a citation? Also, can you clarify what you mean by ‘greatly’, eg increased by 50%? Page 12, Lines 30-34: The authors write “Assuming a constrained diagnostic testing budget and a cost of \$5 per rapid diagnostic test and of \$30 per PCR laboratory diagnosis, we calculate that household (rapid antigen) testing would detect four times more cases than laboratory (PCR) testing, despite its lower sensitivity, under a range of prevalence levels (from <0.05 to >0.4).” The math leading to this conclusion isn’t clear – can the authors describe in greater detail how they arrived at this statement and/or provide a diagram showing the costs and tradeoffs? Discussion: The Discussion could be pared down somewhat to focus on the most salient points. I think the authors should also discuss the lack of additional impact (on a public health level) of testing household contacts. The modest increase in sensitivity (if there is any increase at all – one could argue that sensitivity of 0.68 and 0.71 are essentially the same, especially for a relatively imprecise measurement in a modest-sized sample of positive tests) does not
--	--

likely justify the additional cost of testing household contacts. Thus, in my mind, this is a very helpful study to show that index case testing is critically important, and that governments and institutions should focus on testing these cases with rapid testing and not spend additional \$\$ testing other household members. This would likely save time and money (though knowing for certain might require a cost-effectiveness analysis beyond the scope of this paper). As the manuscript and discussion are currently framed, the authors have taken a different interpretation, which seems to support testing household contacts despite the minimal improvement in sensitivity. Based on their results, I disagree with this proposed approach, and instead suggest that there is evidence to support deploying HCWs to test index cases only, using rapid antigen tests.

Page 13 Lines 7-11: The authors write “We ... found that testing additional household members increased the sensitivity of detection of COVID-19 infection without compromising specificity by much.” This statement contradicts the Results, where the authors noted that testing additional household contacts led to a “large drop in specificity” (Page 12, Line 10). These statements need to be reconciled and clarified. Furthermore, this statement notes that testing additional household members increased sensitivity. As I noted above, a change from sensitivity of 0.68 to 0.71 is not likely meaningful, and comes at increased cost. The increase in sensitivity of saliva testing was greater (because the sensitivity was initially so low for testing a singled family member) but is also not likely to be meaningful, because nasal testing should be used instead of saliva testing, based on these results.

Pages 13-14: It should be noted that potential participants who were afebrile were excluded from the study, and the authors should discuss how this may impact their results. If afebrile patients had high viral loads, they could potentially be transmitting virus as well, but we do not have any guidance from this study on how antigen testing would perform in this population.

Page 14, Lines 19-20: The authors write “Our reports of higher sensitivity when testing additional household members is similar to the interpretation of flock-based testing in animals.” Please see my comments above about stating higher sensitivity – I do not think the results provided suggest there is clinically meaningful higher sensitivity in testing household contacts compared to testing only the index case. I think the Discussion should be re-worked with this in mind.

Page 15, Lines 20-26: This study is helpful in describing a household-based testing method and noting that it can be successfully implemented by CHWs. However, there is no evidence of scalability here, nor is it clear whether COVID-19 is a good model for other infectious diseases that CHWs could test for on a household level. This may merit additional discussion and scaling back on the statements about scalability and proof-of-concept. There are few limitations described in the Discussion. The authors can consider adding limitations around reproducibility (if CHWs in different areas may train/operate differently), how pre-test probability affects test performance and the likelihood of a positive test result being a false vs true positive (and, similarly, a negative test result), how direct-to-consumer marketing/sale of antigen tests may change the role of CHWs, etc.

VERSION 1 – AUTHOR RESPONSE

Reviewer: 1

Dr. Isah Abubakar Aliyu, Bayero University

Comments to the Author:

The research work by Sania et al. titled Rapid antigen testing by community health workers to improve detection of SARA-CoV-2 in Dhaka, Bangladesh submitted to BMJ Open, aimed at evaluating the diagnostic performance and feasibility of rapid antigen testing for SARS-CoV-2 detection by community support team in low-income communities. Though the paper was well written and authors attempt to address the objectives of their research, i have the following observation to make the paper better.

1) Methods; it was not clear how the household were selected for their research, how these household members were mobilized, it was also not clear which part of Dhaka was sampled. they did mention low income communities only.

Response: The study participants lived in Dhaka South and North City Corporations (Page 7, Paragraph 2). The community health workers were already working in these areas providing COVID-19 support for members of these low-income communities as part of the community support teams. These community volunteers were recruited by BRAC or were members of several civil society organizations (i.e. Youth Platform, Utshargo Foundation, Himu Paribahan, CDP, Platform, and Young Bangla). As part of their activities, they went door to door to identify potential COVID-19 cases based on self-reported symptoms. When symptomatic individuals were identified they were referred for testing and advised on isolation.

For the purpose of this study, we trained a subset of these community health workers to conduct rapid antigen testing who then recruited these individuals during their daily work going door to door, until the calculated sample size (number of households to be tested) was met. We have now added the details about the syndromic surveillance program of the community support teams in the methods section to provide this improved context (Page 7, Paragraph 2)

2) Result; on the page 10 of 23, line 8: " Table 1 shows the sex and age distribution of the participants...." this statement sound inappropriate, it is their result rather then the table. It should ideally beour result showed that/the finding of this study showed....etc (Table 1). [NOTE FROM THE EDITORS: please feel free to rebut this reviewer suggestion]

*The tables should be presented in a more scientific manner.

Response: If the manuscript is accepted for publication, we will reformat the tables to meet the criteria set by the Journal

*Page 10 of 23 line 36,; same result was presented in table and figure, the authors should chose either to represent same group of data.

Response: The reviewer is correct that the same data is presented on the diagnostic performance of the rapid antigen tests in comparison to RT-PCR in both the table and the figure. However, the way these resulted are aggregated (all household index cases in the Table, and index cases who were all symptomatic, as well as with the addition of 1 or 2 household members for the figure) does translate to different analytical results. Our reason for presenting in this way is for ease of interpretation - the figure aiming to show the incremental improvement (or not) with additional household members, and the table to show the more standard and straightforward table of sensitivity and specificity as well as

PPV, NPV and Accuracy (that are not reported in the figure). We have tried to make our reasoning clearer in the revision.

*Conclusion; the bases with which the authors recommend using community health worker to do laboratory testing was not justified by this research.

Response: Our results support that sensitivity and specificity of rapid antigen tests administered by community health workers are comparable to the performance of rapid antigen tests in other settings. Testing COVID-19 with rapid antigen tests does not require any specialized laboratory settings or highly skilled medical personnel. In fact, antigen tests are now being used as at-home tests in many countries.

While the RATs are not as sensitive as RT-PCR, because of their low cost and the ease of administration mass testing coverage can be achieved by allowing community health workers to administer these tests in countries that are constrained by limited diagnostic and healthcare infrastructure. We believe that the evidence we present, both sample collection and rapid testing in households that otherwise do not have access to diagnostic testing, as well as just sample collection by community health workers for laboratory testing using RT-PCR are backed up. Sample collection by these community health workers has actually been adopted by IEDCR since the study ended, precisely because it saved resources and reduced the workload of their laboratory staff (who previously had also had to collect samples). We have included additional information about the willingness of people to be tested to further justify this point. This was likely because the approach of sending laboratory staff to communities in full PPE generated a lot of stigma, rather than more discrete testing by trusted community health workers. We hope this helps to address the reviewers' concern.

Reviewer: 2

Dr. Lisa Bebell, Massachusetts General Hospital

Comments to the Author:

The authors present their findings on an important research topic – testing for SARS-CoV-2 by community support teams in Bangladesh. This is an important study to help inform how rapid testing can expand access to COVID-19 diagnostics, and support the role of CHWs. However, I think that prior to publishing this work, additional detail is needed in the Methods and Results sections. Furthermore, I disagree with the interpretation of the change in sensitivity and specificity that occurred when additional household members were tested. I think these results should be reconsidered and the Discussion section reframed.

Response: Thank you for your positive feedback. We have edited the methods and results according to your feedback below and further discussed the results regarding sensitivity and specificity with testing of additional household members.

Specific comments:

Abstract:

I recommend updating/reframing the abstract to reflect my comments below for each individual section.

1. Introduction:

Appropriate to the topic and flows well. The introduction could be shortened slightly if word count is of concern – especially the last paragraph, which discusses healthcare worker deployment and testing in greater detail than is necessary in the introduction.

Response: The description of the Community Support Teams program intended to provide the readers a broad overview of the context of the study. As there is not a constraint of word count, we kept the introduction unaltered.

2. Methods:

2.1 Additional clarity is needed in the methods section to help the reader understand exactly how CHWs and participants were selected, and what samples were collected from each person for each type of test – the results section should also be updated to parallel what is delineated in the methods.

Page 8, Lines 20-24: Please clarify how 'participants' were chosen. There is some clarification provided in the following sentences (noting that CHWs were identifying symptomatic cases) – were all participants suspected cases? How were the smaller CHW groups (n=60, 30 teams) selected? Did they operate in a single geographic area, or in all communities?

Response: The community health workers who undertook sample collection and rapid antigen testing (n=60, 30 teams) were selected by the Area Managers (line managers coordinating their operations) of Community Support Teams based on their performance, with additional consideration to minimize their travel time and maximize geographical coverage (Page 7, Paragraph 2). At the time of the study, 645 volunteers worked across Dhaka South and Dhaka North City Corporations, serving the poorest communities. The community health workers were trained to clinically assess individuals reporting COVID-19 symptoms, through door-to-door household visits. Upon screening, guided using a pre-programmed mobile phone-based application, they counseled individuals identified as likely COVID cases to maintain home isolation and quarantine of their entire household for 14 days and provided support throughout the quarantine period, including essential medicines, facilitating food support to vulnerable households and telemedicine / hospital referrals for those progressing towards severe illness. Only suspected cases, identified based on their showing symptoms, participated in the study.

2.2 Page 8, Lines 26-27: The authors note “Participants aged 16 years or older with no previous medical history of bleeding disorders were considered eligible for the study.” Do the authors mean community members? How would they be participants if they hadn't already been consented for the study? Or were they already consented for another study? Why were people with bleeding disorders excluded?

Response: We replaced participants with community members. People with bleeding disorder were excluded because of the concern of potential injury during nasal swabs. This concern was raised during the ethical and scientific review of the proposal.

2.3 Were household members only eligible if they were also over 16 years of age?

Response: additional household members were only considered eligible if they were 16 years or older. (Page 7, paragraph 2)

2.4 Page 8, Line 46: The authors write “nasal swabs from each nostril” – does this mean one swab from each nostril? Were these swabs of the anterior nares? Some of these details are important for correctly interpreting the results.

Response: Two nasal swabs were taken from both nostrils. Swabs were taken from the mid-turbinate. The rapid test sample contained one nasal swab, whilst the sample for PCR contained the left nasal swab combined with the OP swab (see below).

We have clarified this in the manuscript (Page 8, paragraph 1)

2.5 Page 8, Line 47: When the authors write “One nasal and OP swab was pooled for RT-PCR” do they mean that the nasal and OP swab were combined to create one sample for testing?

Response: Yes, the nasal and OP swabs were combined to create one sample for PCR testing, as this was the standard guideline followed at IEDCR. Now clarified in the text (Page 8, paragraph 1)

2.6 Page 8, Lines 51-52: The authors write “The nasal and saliva samples were tested independently with rapid antigen tests” Does this mean that no OP swabs were tested with rapid antigen tests? And that each nare was tested separately with a rapid antigen test for each nare?

Response: No OP swabs were tested by rapid antigen tests. Only nasal and saliva swabs were tested by rapid antigen tests. In the pilot phase of the study, we evaluated the performance of OP, nasal and saliva swabs among symptomatic cases, which showed the performance of nasal swabs was slightly inferior to OP swabs. There were concerns of injury during OP swabs collected by the CHWs, so the rapid antigen tests were only performed on nasal and saliva swabs. (Page 8, paragraph 1)

2.7 Page 8-9: The authors write “The manufacturer reports that the STANDARD Q COVID-19 Ag Nasal Home Test has a sensitivity of 94.94% (75/79) and specificity of 100% (217/217)(16) and that the STANDARD Q COVID-19 Ag Saliva Home Test has a sensitivity of 94.74% (18/19) and a specificity of 100% (73/73).(17)” Was this test truly validated on this few (<300) samples? Are there real-world results reporting the field sensitivity and specificity? I ask because the reported sensitivity and specificity seem to high to be realistic.

Response: The manufacturer reports of sensitivity and specificity are likely higher because they are tested on laboratory samples. Higher sensitivity in laboratory settings is probably due to higher viral load. We mentioned the real-world performance in the discussion (page 12, paragraph 3). We also included in the method section that the sensitivity and specificity is lower in real world settings (page 8, paragraph 2). The sensitivity and specificity of the STANDARD Q COVID-19 Ag Nasal Home Test can be found here: <https://www.sdbiosensor.com/product/main?bcode=11&bcode=COVID-19%20Products>. The sensitivity and specificity of the STANDARD Q COVID-19 Ag Saliva Home Test can be found here: https://www.sdbiosensor.com/product/product_view?product_no=236.

2.8 Page 9, Lines 14-21: This paragraph describes patient involvement. Who were the patients? Were they the same as the participants? I find this confusing – please clarify whether this describes participant involvement or patients who were not study participants. Also, how were patients and their family members involved with the study design?

Response: The journal requires to report patient and participant involvement. We included patients in the section heading (as per the journal guidance) but replaced patients with participants in the section text. During the design of the study we consulted the community members regarding their preference on testing location, at their home or nearby community centers, to inform the overall study design. (Page 8, paragraph 4)

2.9 Page 9, Lines 31-34: Do the authors mean that they studied the potential cost savings of substituting antigen for PCR testing? Please clarify.

Response: We examined the numbers of tests that could be performed for the same budget (which was a major limitation to the numbers of tests performed), and how the resulting numbers of confirmed positive tests would then compare when substituting rapid antigen testing for PCR testing, under a specified incidence in the community. We have now clarified these details in the methods.

3. Results:

This section does not parallel the description in the Methods section. Additional detail in the Methods section will help – the Results should then describe all the tests discussed in the Methods section. As it is written now, the Results section does not mirror the Methods section well, especially when describing the pooled versus non-pooled testing results and nasal versus OP versus saliva testing results.

Response: We apologize for the brevity of the statistical analyses section of the methods. Our results section was more clearly laid out and we have now reordered the statistical analysis section of the methods, adding detail to correspond directly to and mirror the results section.

3. 1 Page 10, Lines 5-6: How many potential participants declined to participate?

Response: Participants were only recruited into the study if they consented to be tested, which we now clarify in the manuscript (page 7, paragraph 3). As a result, we unfortunately do not know how many participants were considered eligible for recruitment but declined testing. However, we did monitor 'willingness to be tested' throughout the pandemic as part of the Community Support Team intervention. At the time of the study, this indicator increased from just under 25% to 50% of all symptomatic people identified by (all) community health workers i.e. not just the subset collecting samples for the study. We are unfortunately not able to disaggregate willingness to be tested amongst those that the community health workers in the study tried to recruit, but we have no reason to believe this would be particularly different from the average. The reason for the increase in willingness to be tested is likely due to a number of reasons, including tests being directly offered (they were previously not available, or were only being tested at IEDCR therefore results would not be returned till 1-2 days later), and increasing awareness of access to services when testing positive, as well as the fear of the growing epidemic wave. These details are being explored in more detail in an overall evaluation of the whole Community Support Team programme that we are drafting for publication.

3. 2 Please clarify the results reporting saliva samples – are these saliva only, or OP samples, or pooled samples?

Response: We clarified this in the result section (page 10, paragraph 2)

3.3 Page 11, Lines 43-45: This sentence seems like it would be better placed in the Discussion.

Response: We moved this sentence to the discussion included in the paragraph describing the limitations of the study (page 14, paragraph 3)

3.4 Page 11: The authors begin a section with the heading "Testing additional household members yields better rapid antigen test performance:" However, this paragraph describes essentially no change in sensitivity of nasal testing by adding 1-2 household contacts (an increase of 0.68 to 0.71 does not seem meaningful) and a decrement in specificity for both nasal and saliva test performance. I would not call this 'better rapid antigen test performance'. Thus, I recommend re-heading this section, perhaps with "Testing additional household members decreases rapid antigen test specificity" to ensure the message is clear.

Response: We changed the heading to, "Testing additional household members only marginally increases sensitivity but decreases specificity".

3.5 Page 12, Lines 16-19: "Using the rapid antigen testing approach, whereby individuals were tested in their households and immediately given their results,

willingness to be tested greatly increased.” I believe this statement is true – can the authors provide a citation? Also, can you clarify what you mean by ‘greatly’, e.g. increased by 50%?

Response: We speculate that the increase in testing volume in these communities was a result of household level testing provided through this study, and we further infer this through the self-reported experiences of the community health workers. We added a sentence in the results to quantify this change and improve clarity (page 12 paragraph 2). There are confounding factors and some room for how the changes in testing could be interpreted which was elaborate more on in the discussion.

3.6 Page 12, Lines 30-34: The authors write “Assuming a constrained diagnostic testing budget and a cost of \$5 per rapid diagnostic test and of \$30 per PCR laboratory diagnosis, we calculate that household (rapid antigen) testing would detect four times more cases than laboratory (PCR) testing, despite its lower sensitivity, under a range of prevalence levels (from <0.05 to >0.4).” The math leading to this conclusion isn’t clear – can the authors describe in greater detail how they arrived at this statement and/or provide a diagram showing the costs and tradeoffs?

Response: Apologies for the lack of detail on this point. We had been aiming for brevity and did not intend to be opaque. We have added a Figure 4 to illustrate this point which we hope is clearer in our revision.

4. Discussion:

The Discussion could be pared down somewhat to focus on the most salient points. I think the authors should also discuss the lack of additional impact (on a public health level) of testing household contacts. The modest increase in sensitivity (if there is any increase at all – one could argue that sensitivity of 0.68 and 0.71 are essentially the same, especially for a relatively imprecise measurement in a modest-sized sample of positive tests) does not likely justify the additional cost of testing household contacts. Thus, in my mind, this is a very helpful study to show that index case testing is critically important, and that governments and institutions should focus on testing these cases with rapid testing and not spend additional \$\$ testing other household members. This would likely save time and money (though knowing for certain might require a cost-effectiveness analysis beyond the scope of this paper). As the manuscript and discussion are currently framed, the authors have taken a different interpretation, which seems to support testing household contacts despite the minimal improvement in sensitivity. Based on their results, I disagree with this proposed approach, and instead suggest that there is evidence to support deploying HCWs to test index cases only, using rapid antigen tests.

Response: This is an excellent point and we have revised the discussion to incorporate this as a clear recommendation.

4. 1 Page 13 Lines 7-11: The authors write “We ... found that testing additional household members increased the sensitivity of detection of COVID-19 infection without compromising specificity by much.” This statement contradicts the Results, where the authors noted that testing additional household contacts led to a “large drop in specificity” (Page 12, Line 10). These statements need to be reconciled and clarified. Furthermore, this statement notes that testing additional household members increased sensitivity. As I noted above, a change from sensitivity of 0.68 to 0.71 is not likely meaningful and comes at increased cost. The increase in sensitivity of saliva testing was greater (because the sensitivity was initially so low for testing a singled family member) but is also not likely to be meaningful, because nasal testing should be used instead of saliva testing, based on these results.

Response: We edited the sentence to “We also found that testing additional household members increased the sensitivity of detection of COVID-19 infection with a corresponding decrease in

specificity". The statement on "large drop of specificity" apply to saliva samples and combined data analysis including results from nasal and saliva samples (please see figure 1). We also edited the discussion section to better reconcile these differences and to provide more targeted recommendations, indicating that the improved sensitivity was marginal following the reviewers points above.

4.2 Pages 13-14: It should be noted that potential participants who were afebrile were excluded from the study, and the authors should discuss how this may impact their results. If afebrile patients had high viral loads, they could potentially be transmitting virus as well, but we do not have any guidance from this study on how antigen testing would perform in this population.

Response: We have now mentioned this as a limitation in the discussion section. (Page 14, paragraph 2)

Page 14, Lines 19-20: The authors write "Our reports of higher sensitivity when testing additional household members is similar to the interpretation of flock-based testing in animals." Please see my comments above about stating higher sensitivity – I do not think the results provided suggest there is clinically meaningful higher sensitivity in testing household contacts compared to testing only the index case. I think the Discussion should be re-worked with this in mind.

Response: We revised page 14, paragraph 2 to reflect this suggestion.

Page 15, Lines 20-26: This study is helpful in describing a household-based testing method and noting that it can be successfully implemented by CHWs. However, there is no evidence of scalability here, nor is it clear whether COVID-19 is a good model for other infectious diseases that CHWs could test for on a household level. This may merit additional discussion and scaling back on the statements about scalability and proof-of-concept.

Response: We agree that in our enthusiasm we overstated the points about scalability. However, there was considerable reluctance within the government laboratory sector to allow testing or sample collection to be carried out in the community. This had major ramifications, limiting the tests performed and diagnoses made (see Figure 4), being very expensive per test and logistically difficult, not being acceptable in communities (see points above about willingness to be tested) and reducing access to care (for which a positive test was required for admission to COVID-19-serving facilities and for which a negative test was required for non-COVID-19 facilities). This was therefore also a further cause of inequity in these already vulnerable communities. Our study influenced practice in Bangladesh, as community health workers, whilst trusted by communities were not trusted by medical scientists running the laboratories. However the results from this study were acceptable and championed into policy by the scientists who had previously been disapproving. We obviously have a lot to say on this matter for which there is not sufficient scope in this manuscript (but we cover this more in the evaluation manuscript mentioned above that we are currently drafting). We have therefore curbed our enthusiasm and present less emphatic but, we believe, justified statements.

There are few limitations described in the Discussion. The authors can consider adding limitations around reproducibility (if CHWs in different areas may train/operate differently), how pre-test probability affects test performance and the likelihood of a positive test result being a false vs true positive (and, similarly, a negative test result), how direct-to-consumer marketing/sale of antigen tests may change the role of CHWs, etc.

Response: Thank you for these suggestions. We elaborated on these suggestions in the updated limitation section. (Page 14, paragraph 3)

VERSION 2 – REVIEW

REVIEWER	Lisa Bebell Massachusetts General Hospital
REVIEW RETURNED	09-May-2022

GENERAL COMMENTS	This is a revised version of a manuscript I previously reviewed. In their comments, the authors thoughtfully responded to my comments, and those of the other reviewer and editor(s). I found their explanations to be both enlightening and appropriate, and the manuscript revisions were carried out well. I have no further comments or suggestions, except to congratulate the research team on carrying out an important study that I hope will provide evidence for community-based testing for COVID-19 and other infectious diseases in the future.
--